# Deciphering the COVID-19 Health Economic Dilemma (HED): A Scoping Review

**DOI:** 10.3390/ijerph18189555

**Published:** 2021-09-10

**Authors:** Arielle Kaim, Tuvia Gering, Amiram Moshaiov, Bruria Adini

**Affiliations:** 1Department of Emergency and Disaster Management, Faculty of Medicine, School of Public Health, Sackler Tel Aviv University, P.O. Box 39040, Tel Aviv 6139001, Israel; ariellekaim@mail.tau.ac.il (A.K.); tgering@gmail.com (T.G.); 2Israel National Center for Trauma & Emergency Medicine Research, The Gertner Institute for Epidemiology and Health Policy Research, Sheba Medical Center, Ramat-Gan 5266202, Israel; 3School of Mechanical Engineering, Iby and Aladar Fleischman Faculty of Engineering, Tel Aviv University, Tel Aviv 6997801, Israel; moshaiov@tauex.tau.ac.il

**Keywords:** health, economics, COVID-19, decision-making, review

## Abstract

Lessons learnt from the initial stages of the COVID-19 outbreak indicate the need for a more coordinated economic and public health response. While social distancing has been shown to be effective as a non-pharmaceutical intervention (NPI) measure to mitigate the spread of COVID-19, the economic costs have been substantial. Insights combining epidemiological and economic data provide new theoretical predictions that can be used to better understand the health economy tradeoffs. This literature review aims to elucidate perspectives to assist policy implementation related to the management of the ongoing and impending outbreaks regarding the Health Economic Dilemma (HED). This review unveiled the need for information-based decision-support systems which will combine pandemic spread modelling and control, with economic models. It is expected that the current review will not only support policy makers but will also provide researchers on the development of related decision-support-systems with comprehensive information on the various aspects of the HED.

## 1. Introduction

The emergence of the novel Coronavirus disease (COVID-19) in December of 2019 in Wuhan, China has rapidly left a significant dent on the global community. As of 9 December 2020, the pandemic has impacted 191 countries with over 68 million global cases confirmed and over 1.5 million deaths [1]. This novel and emerging illness was declared by the World Health Organization (WHO) a Public Health Emergency of International Concern (PHEIC) on the 30th of January 2020 [2]. The global crisis presented governments and decision makers with a myriad of public health, political, economic, societal and cultural dilemmas [3]. 

The high virulence and transmissibility of the virus, and the lack of effective drugs, antivirals and vaccinations threatened the inundation of health systems in heavily stricken regions and made “flattening-the-curve” of infections a top priority [3,4,5]. The principal intent of flattening the epidemic curve is to decrease morbidity and delay infections in order to diminish the health impact by spreading the impending burden on hospitals and infrastructure over time [6]. With this priority in mind, non-pharmaceutical interventions (NPIs) were invoked, most commonly being the enforcement of mass or targeted quarantines, restrictions on mass gatherings and closure of workplaces as well as religious, recreational, educational and other non-essential services and facilities [7]. Additional NPIs included travel bans, compulsory wearing of facemasks, stressing hygiene and disinfection, culling of infected animals and plants, risk communication, enforcing of social distancing and isolation of active cases in designated areas and the protection of high-risk populations [8]. Prior to the COVID-19 outbreak, the effectiveness of NPIs had primarily been studied theoretically, particularly through the context of the pandemic influenza [9,10,11]. 

The implementation of interventions to “flatten the curve” during the ongoing pandemic characterizes the high societal prioritization of public health and its primary focus during the COVID-19 outbreak among most world leaders and decision makers [12,13]. These directed efforts reflect the collective norms for preservation and protection of human life, with reduction of mortality rates standing as a token of success [12,14]. Furthermore, the presented challenge of preventing a healthcare system collapse was stressed, as several medical facilities in countries such as Italy and Brazil rapidly approached collapse due to inadequate infrastructural preparedness and failed management and implementation by the authorities [15,16]. 

For the World Health Organization (WHO), a key global player in public health outbreak response, the declaration of a PHEIC during the COVID-19 outbreak presented an opportunity to identify what is subject to mandatory reporting by WHO country members as part of the International Health Regulations (IHR) revisions that were adopted in 2015 [17]. Through the setting of normative and technical standards, the WHO asserts its public health role and power [17]. As governments began navigating the uncertain territory of the novel outbreak, it was not immediately clear how it originated, how people could protect themselves, what the actual scope of the outbreak was and how it would be contained [18]. Misinformation and disinformation, politicization and denial of science unfolded alongside the spread of the virus [19]. These principal elements contributed to the accented prioritization of the health domain during the management of the COVID-19 outbreak. 

As extraordinary mitigation measures were directed by decision-makers, the intensity and far-reaching impacts on additional societal domains of life and segments of society became apparent rapidly [8,20,21]. The pandemic did not only present a PHEIC but rather also presented a societal, economic and political event whose impacts are far-reaching and long-term [21,22]. The collateral new reality that COVID-19 has generated exposed the fragility of our modern systems, including the social inequities among vulnerable populations and the complexity of addressing the scale and scope of the varied domains affected as a result of the delicate nature of this crisis [22,23]. 

Amidst the impacts of the disease on mortality and morbidity, the global economy has also encountered substantial adverse disruption. To date, the outbreak has occasioned the largest downturn since the financial crisis of 2008 and a general downward trend in the global economy triggered the worst global recession since World War II [24]. The International Monetary Fund’s (IMF) global growth contraction in 2020 is estimated to be a volatile negative 3.5% [25] and according to the United Nations University, the COVID-19 crisis may increase global poverty levels by 420 to 580 million people [26]. The crisis has occasioned vast increases in unemployment [27], extensive reduction in flow of goods through global supply chains [28], widespread cutback in service availability [29], severe resource scarcities and depletion [30] and major oil and stock market shocks [31].

In light of this, of a particular interest for this systematic review is the consequential intersection between the public health and economic domains as well as the underlying costs to each domain which are critically contingent on the measures taken. For decision makers, precisely the aforementioned interaction, or what is hereby referred to as the health-economy dilemma (HED), has proven to be a substantial challenge. The nature of the debated tradeoff of the HED examines both local and global issues. For example, the effect of NPIs (for example, a lockdown) on infectivity levels versus the incapacity of households to maintain their income. On a global scale, the HED touches on issues such as the impact of international border closures on curbing spread of the virus versus its influence on service-oriented economies (particularly those that are largely reliant on tourism) such as Greece, Portugal and Spain [32].

While inaction was not a feasible option for all but few authoritarian countries, many governments deliberated the optimal strategy for containment of the virus. Opportunities for lessons learned from past similar events have been constrained by the low incidence of global pandemics in history and the novel challenges posed by an increasingly globalized world. In addition, the endless combinations of NPIs, the distinct circumstances of each country and the complexity of pattern predictions of a novel pathogen, all added to the difficulty of the task. Countries struggled to strike a balance between the two domains, attempting to simultaneously flatten the infection curve and the “macroeconomic crisis curve” [33].

The COVID-19 outbreak and its consequent economic implications have driven a surge of peer-reviewed publications and gray literature by scholars, think tanks, policy institutions and international organizations, integrating economic theory with epidemiological models. In order to better understand the interaction between these two domains during the COVID-19 outbreak, we aim to provide an extensive mapping of consolidated literature on this subject. Such a review, in the context of the COVID-19 outbreak, will inform decision makers on issues concerning the optimization of NPIs in view of the HED and its related tradeoffs. 

## 2. Review Methodology

The literature review was conducted systematically in a multi-step process (Figure 1). English language records were searched in Scopus, PubMed and Google Scholar utilizing the following search string *(health AND economy) AND (COVID-19 OR coronavirus OR SARS-CoV-2)*. Only empirical papers dating from 1 December 2019 until 12 December 2020 were included. Following this, initial screening of results was conducted based on the title of publication and removal of duplicates from different databases. Abstract review was further used to remove irrelevant records that did not focus on the health economic dilemma (HED), later accompanied by a full text review, for a final removal of publications that did not elucidate elements of the dilemma. The remaining articles were included in the review to identify the main themes and insights concerning the HED. 

The papers that completed full review (*n* = 50) underwent categorization of main themes and messages. This helped to map the overall landscape of the literature as is referred to in the primary body of the text. It also unveiled the need for information-based intelligent decision-support-systems that combine health and economy considerations. The study did not collect any primary data, and no data was collected from human subjects directly by the authors. 

## 3. Results

### 3.1. The Nature of the Tradeoff

The two-dimensional nature of the pandemic has polarized the COVID-19 debate landscape, with arguments from decision makers and analysts ranging from either “Whatever it Takes, Grandma or the Economy?” [34,35]. The discussions about the perceived “trade-off” of the HED resonate as a heavily socio-politically polarizing issue and induce societal tension. The restricted training of health care practitioners and epidemiologists in economic concerns, as well as economists in elements related to epidemiology, has intensified this polarization, restricting the range of decision-making tools [36]. To aid policy implementation relevant to the management of the ongoing pandemic and potential outbreaks of similar nature, this literature review offers new policy insights. 

The themes found primarily discussed throughout the literature on the subject include: the general public’s views and priorities when considering the HED; the public health costs of an unchecked epidemic; social distancing as an effective strategy for improving public health metrics and reduction of medical and potential economic costs; a debate over whether social distancing measures are justified given the economic toll; the impact of social distancing policies on the economy; cross-country comparison of various policy approaches; the need for a more nuanced approach and the role of modeling to achieve this; and lastly, an example of a proposed viable policy intervention (See Table 1). To clarify, when referring to social distancing throughout the review, we refer to the measures intended to prevent the spread of a contagious agent between people and reduce the number of times people come into close contact with each other.

### 3.2. General Public Views and Tradeoff Priorities

The unique aspects of COVID-19 raise questions about where the general public’s support lies when it comes to this dilemma. When the Dutch and American public were probed with the question of what they value more, pro-health or pro-economy, in the context of the pandemic, an overwhelming majority of participants supported both dimensions [13,37]. In contrast, a study conducted in China which assessed whether the public cared more about containing the virus or sustaining normal economic activities during the pandemic, found that economic expectations are significantly affected by the severity of the pandemic, rather than by the level of economic recovery [38]. Therefore, in accordance with the studies’ findings, containment measures for the virus must first be prioritized, as a lower number of COVID-19 cases significantly correlated to increases in individual expectation for Gross Domestic Product (GDP) growth rate [38]. We continue by elucidating the complexity of the presented issue by first exploring the posed costs to an unchecked COVID-19 outbreak.

### 3.3. Morbidity and Mortality Cost of an Unchecked COVID-19

The human cost of the COVID-19 outbreak is undoubtedly substantial. During the early stages of the outbreak, upper end estimates of projected deaths, in the absence of effective protective measures or vaccination, reached as far as 6% of the global population (half a billion deaths) [39]. Gourinchas (2020) [33] provided estimates of an unrestrained COVID-19 pandemic: Given a 2% case fatality rate (CFR) baseline, overwhelmed health systems and 50% of the world population infected, 1% of the world population—approximately 76 million people—would die. Aside from the high levels of morbidity and mortality that would be incurred with a “laissez-faire” approach, the researchers further elucidated the risk of potential health system inundation [33]. This is explained by the fact that given a limited capacity of any health care system, such as finite capacity of intensive care units, hospital beds, skilled health professionals, ventilators, etc., there is a threshold for how many patients can be handled at a given point in time [33,39,40]. Grech (2020) [39] determines that hospitals may just collapse, and basic provision of care may fail, whereas for even non-novel conditions or procedures (such as childbirth) where established treatments exist, morbidity and mortality will be incurred. The literature elucidates that bottlenecks were created in the healthcare systems of countries such as China and Italy which necessitated what resembled a “wartime triage” to diffuse these conditions [33,39,40,41]. Medical professionals were forced to decide who would receive life-saving care and who would be left without the necessary care to survive the disease [40,41]. 

The key issues of mitigation thus for epidemiologists, as presented by Anderson et al. (2020) [4], are minimizing morbidity and associated mortality, avoiding epidemic peaks that overwhelm health-care services and flattening the epidemic curve in waiting for vaccine development and at-scale manufacturing.

### 3.4. Is Social Distancing an Effective Strategy for Improving Public Health Metrics and Reducing Medical and Potential Economic Costs?

To mitigate the infection and death curve and to ensure a more manageable stream of those requiring care from the healthcare system, aggressive social distancing has been deemed a viable and necessary NPI strategy by many experts [33,41,42,43,44,45]. The internal and external benefits of social distancing stem from reducing person-to-person contacts which facilitate a lower likelihood of getting sick and potentially transmitting the illness [42]. Chen et al. (2020) [40] establish the importance of social distancing as a mitigation strategy by modeling the medical costs of keeping the US economy open during the COVID-19 outbreak. The findings conclude that the unmitigated medical costs of the first wave account to over one trillion dollars (approximately 5% of the US economy) and could substantially be brought down to just 35$ billion dollars with social distancing alone. Similarly, Brzezinski et al. (2020) [45] conducted assessments of medical and economic costs associated with COVID-19 and concluded that in a non-lockdown scenario, the cost will be 16.1% of annual GDP per capita, while 15.2% of annual GDP will be affected if a lockdown is imposed. When the statistical value of life was taken into account, these values increased to 20.9% and 18.1% of annual GDP per capita respectively [45]. These findings suggest that economic costs are inevitable even in non-lockdown scenarios, with economic performance barely improving while substantial increases in medical costs are incurred. Consistent with both sets of findings, Silva et al. (2020) [43] analyzed seven different social distancing scenarios which resulted in varying epidemiological and economic consequences, with the simulations supporting the claims that lockdowns are necessary for controlling the number of infected and deaths and that economic losses are unavoidable. 

Where aggressive policies of social distancing were implemented, strong results in reduction of the virus spread were seen such as in China, Taiwan and Singapore [33]. However, juxtaposing claims were made by Stock (2020) [46] and Lin and Meissner (2020) [47] with regards to the United States, where they indicated that despite the implementation of social distancing regulations, weekly deaths and growth of COVID-19 cases did not slow substantially.

### 3.5. Is Social Distancing Justified given the Economic Toll?

Substantial amounts of literature acknowledge the detrimental and vast impact of the outbreak on economic activities [33,34,44,45,48,49,50,51,52,53,54,55,56,57,58,59,60,61,62,63,64,65]. While it is well accepted that the economic perils of a pandemic are not trivial, less agreement in the literature exists with regard to whether the consequences of various NPIs, such as social distancing, are appropriate. McKee and Stuckler (2020) [54] point out that the economic decline itself has an adverse effect on health, while Dorn et al. (2020) [63] suggest that the underlying situation is more complex. They conclude that a larger disease burden would result from non-containment of the pandemic, inherently inducing adverse effects on the economy through the form of reduced trust of consumers and investors, and new infection waves would result in considerable further costs which would not reconcile economic objectives [63]. Similarly, findings from Australia and Indonesia indicate that better public health outcomes are positively associated with higher levels of social distancing and lower economic costs [64,65]. Pertinent to these discussions are the cost- benefit analyses which are comprised of evaluations of long versus short-term benefits to both public health and the economic domains. The models emphasize that if the government assigns a high value for life, this will opt for serious social distancing measures, such as long lockdowns, thereby saving lives, while at a great cost; conversely, if the government assigns a low value of life, this will ensure a low economic cost but will also involve a large number of deaths [66,67]. As addressed by Miles et al. (2020) [66] with regard to findings from the UK, the real costs of social distancing policies will not be known for many years. However, they estimate that costs of social distancing policies will be 10 times or more the scale of benefits (lives saved according to quality adjusted life years (QALYs)) indicating that such measures are likely not warranted. Similarly, findings by Rowthorn and Maciejowski (2020) [67], modeled the costs and benefits of social distancing and found that strict measures of social distancing may be necessary initially to halt explosive spread of disease; yet, once this aim is achieved it would be a mistake to stick to expensive social distancing policies.

### 3.6. Gaging the Impact on the Global Economy

To further the discussion on the appropriateness of social distancing, the literature extensively explores the impact of social distancing policies on various sectors of the economy. As the global and national virus spread emergencies were declared, restrictions of movement resulted in limitations to economic activity [58]. It is well established that epidemics and pandemics often naturally induce recessions; however, efforts to flatten the epidemiological curve of the virus deliberately exacerbate this reduction [33]. Gourinchas (2020) [33] viewed the modern economy as an interconnected web of parties (e.g., employees, firms, suppliers, consumers, banks, financial intermediates). Using this view, Gourinchas argued that if one of the links between these parties is stranded or ruptured by the disease or containment policies, then the outcome is a cascading chain of disruption [33]. A pertinent global scale example is presented by Sattar et al. (2020) [68] which considers the initial closure of the economy in China as a result of social distancing measures and the impacts of this on the global economy. For a country like China, with the second largest economy in the world, making up approximately 16% of the world GDP and being a major global exporter, the closure of the economy elicited trickling consequences and disequilibrium for other associated economies, as many rely on China for manufacturing output and raw materials [32,68]. As illustrated, the disruption to the functioning of global supply chains significantly impacts global trade. The World Trade Organization estimated that the industry fell up to 32% in 2020 due to the coronavirus pandemic [69]. 

Furthermore, the dislocation of many industries is estimated to increase global unemployment. In a low scenario, with GDP dropping by around 2%, an increase of 5.3 million unemployed is predicted, and in a “high scenario”, with GDP growth reducing by 8%, global unemployment would increase by 24.7 million [70,71]. It is well established that vulnerable populations and social groups, particularly the poor, will be disproportionately impacted by these notable declines [70,71,72,73,74]. The footprint of the pandemic will be experienced most considerably in developing countries, as flattening the epidemiological and recession curves would present a more substantial challenge [64,66]. Developing countries generally have less fiscal space and resources to offset the negative shocks [32,70,71,72]. According to initial analysis on the economic impact of COVID-19, regions such as Africa and Asia-Pacific are projected to absorb most far-reaching shocks [32,51]. 

Despite this, in both developing and developed countries, while still investigating the complete extent of impact, the effect on the economy has been profound. It is painfully evident that the increasingly strict measures imposed by governments will leave a toll on macro and micro levels [65]. The immediate and direct impacts from social distancing policies and closures are a sharp and immediate decline in production of a country, as many factories and businesses close down, resulting in aggregate supply shocks, primarily in non-essential industries [52,64,68]. Increasing evidence also indicates that the pandemic is set to significantly decrease aggregate demand for goods and services [32,65,68]. The sectors that the literature has documented as being most severely contracted include the manufacturing sector, the service sector, international trade, tourism and aviation industry and the education sector [32,42,52,65,68].

### 3.7. Cross-Country Comparison

Naturally, the discourse surrounding social distancing policies and the impacts on the economy and health are considered in reference to various approaches undertaken by many countries. While cultural and additional differences such as socioeconomic status, political climate, etc., must be taken into consideration when doing cross-country comparison, approaches fall on a spectrum between limited containment measures, to drastic lockdowns, with most countries found somewhere in between these extremes [66,75,76,77,78,79,80]. Ostling (2020) [78] discusses these two extremes as Sweden and New Zealand, with Sweden’s model illustrating the former extreme and New Zealand the latter. Sweden’s policy put greater weight on economic considerations than other countries, while New Zealand’s policies prioritized elimination of the virus altogether [66,78]. Mixed evidence exists over how Sweden compared to countries that adopted strict social distancing policies, with some finding that Sweden’s looser restrictions resulted in significantly more people dying when compared to its close neighbors of Norway and Denmark (with similar population density, health care systems and climate) and whilst enjoying a lower economic hit [66]. UK data comparison showed significantly higher death rates as compared to Sweden; however, beginning in June 2020, unlike most other European countries which began to ease measures, Sweden saw an increase in cases [66]. Policy divergence in terms of a “Stringency Index” is also presented by Koyama and Desierto (2020) [77] between Brazil and Argentina, Denmark and Sweden and the United Kingdom and the United States. Lastly, Nyarko et al. (2020) [79] compare African containment policies to styles of containing COVID-19 in Hong Kong, South Korea, Japan, Singapore and Taiwan. While the optimal policy is still unclear and it is not yet understood whether certain governments overreacted or underreacted as compared to others, several papers suggest that clear health and economic advantages exist for approaches which aimed to eliminate the virus quickly while rapidly reopening the economy. This approach was best highlighted by New Zealand [59,78].

### 3.8. The Need for a More Nuanced Approach, the Role of Modeling for Optimization

Given the complexity of societies, the myriad of potential policy interventions and the nature of this multifaceted global threat, a more nuanced approach to the HED is necessary in order to achieve optimal and effective policies which reconcile public health and economic interests. When charting a path forward for grappling with the ongoing pandemic, an optimal approach involves dynamically managing the tradeoffs between health and economic costs simultaneously, rather than solely focusing on health priorities or economic ones [73]. As Gourinchas (2020) [33] concludes, some damage will occur, but it will be optimistically short-lived for both domains. If either domain goes unchecked, the pandemic will quickly overwhelm any health system and fatality rates would surge, while the economy would suffer major ruptures to the complex network of economic linkages that allow for the economy to operate [33]. For this purpose, several studies in the literature present developed models to simulate the COVID-19 pandemic in terms of epidemiological and macroeconomic effects (e.g., [43,59,80]). 

Silva et al. (2020) [43] propose one such model which assesses the impact of various social distancing interventions and other control measures on the number of infected, fatalities and economic losses. Their conclusions indicate that governments that chose to preserve the economy by not using severe isolation policies, reached a situation with a high cost in human lives and still took on economic losses as the social costs ended up negatively impacting the economy [43]. A Pareto front-based model evaluation was used to assess and stratify various national responses to COVID-19, while considering both epidemic trajectories and economic losses [59]. Such a model assumes no a priori objective preference; hence, it exposes the tradeoffs between economic and health performances. Kochańczyk and Lipniacki (2020) [59] conclude from this model that protecting the economy and saving lives are non-trade-off objectives and can be optimized using a “hit hard, hit fast” strategy. Lastly, a case study by Agarwal et al. (2020) [80] analogously used a synthetic interventions method and evaluated different interventions that were used to mitigate the COVID-19 pandemic. The findings of this simulation indicate that moderate mobility restrictions will be sufficient for flattening the curve and diminishing considerable economic impacts. These models attempt to couple and merge dynamics of infection and macroeconomic effects and provide quantitative, tangible guidance about how best to apply NPIs in order to augment public health measures and economic activity.

### 3.9. Proposed Viable Policy Intervention

The COVID-19 virus must be defeated but not at the expense of the economy [56]. Various proposed policy interventions have been recommended to contain COVID-19 without locking down the economy. One such proposition includes achieving herd immunity among low-risk groups (such as younger populations) to keep the economy running, while risk groups would be selectively quarantined [50,59,60]. A model analysis by Shalev-Shwartz and Shashua (2020) [60] of this approach finds that it will succeed in avoiding overwhelming the health system, while simultaneously allowing the economy to not enter a hibernation state. Even without a directed policy, spontaneous, voluntary social distancing has been documented to a degree amongst the general public, indicating that draconian interventions may be disproportionate [47,59]. 

The return to normalcy will only occur through a complex assortment of additional interventions and policies. Testing, tracing and mask wearing have limited directly negative economic impacts and have been shown to substantially reduce spread of infections [49]. Additionally, macroeconomic and development policies will play an important role in limiting socio-economic effects, while preserving the capacity of the economy to recover promptly [36]. The literature presents various additional policies, (e.g., direct wage or income support measures, tax deferrals, reduced interest rates, injecting liquidity, etc.) which have been implemented by national authorities and multilateral entities [36,46,51,53].

Though decision-making during the COVID-19 pandemic is no doubt complicated by the need to consider opposing or at times, even contradictory views regarding antagonistic decisions, this is crucial to achieve an effective management of the situation. For example, deciding whether or not to issue a lockdown, mandate vaccinations or instruct on levels of clinical care in overwhelmed medical institutions, may necessitate a relative preference of the health or the economic consequences and thus impact on the course that will be defined. The benefit of considering these varied views is derived from the following factors: (1) As COVID-19 presents not only a health crisis but also a social and economic one, any decision has to first assess the relative contribution or damage that will be caused to each of these domains; (2) Any decision will substantially affect the lives of all populations, thus priorities need to be defined to ensure a just distribution of the burden; and, (3) To obtain an affective compliance of the public to the measures instructed by the authorities, the different concerns must be considered and applied in the decision-making, to achieve understanding and consensus as wide as possible. 

## 4. Conclusions

Lessons learnt from the initial stages of the COVID-19 outbreak indicate the need for a more coordinated economic and public health response. While social distancing has been shown in theoretical and epidemiological literature to be effective as an NPI measure to mitigate the spread of COVID-19, the economic costs have been substantial. New insights combining epidemiological and economic data are emerging which provide new theoretical predictions that can be used to better understand the health/economy tradeoffs. Findings from this literature review aim to elucidate perspectives to assist policy implementation related to the management of the ongoing outbreak and impending outbreaks of similar nature. 

While focusing on a survey of empirical papers, the work towards this review unveiled the need for information-based decision-support systems which will combine pandemic spread modelling and control, with economic models. Currently, as apparent from reviews such as in Musulin et al. (2021) [81] and Tseng et al. (2020) [82], there is a vast volume of studies on virus spread prediction and control models; however, there is a lack of studies that combine such models with economic models to reveal the HED related tradeoffs. Some initial research attempts towards this goal can be found in recent works such as in Yousefpour et al. (2020) [83], Salgotra et al. (2021) [84] and Miikkulainen et al. (2021) [85]. It is expected that the current review will not only support policy makers but will also provide researchers on the development of related decision-support-systems with comprehensive information on the various aspects of the HED. 

Future research inquiries should dedicate an examination to geographic specific COVID-19 impact and respective attitudes regarding the HED. Additional considerations that were not elucidated in the above literature on the HED are the potential increased healthcare expenditures due to lack of attention to non-COVID-19 pathologies as well as the economic costs of productive age loss. Future studies should continue to follow the discussion on the COVID-19 HED in order to deepen the global community’s understanding on various aspects of the subject.

## Figures and Tables

**Figure 1 ijerph-18-09555-f001:**
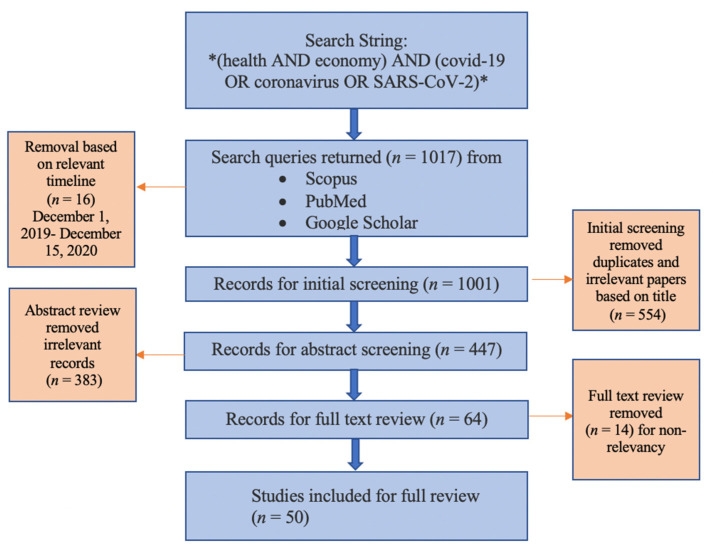
Method flow chart.

**Table 1 ijerph-18-09555-t001:** Health Economic Dilemma (HED) thematic categorization.

Section No.	Theme	Description	Reference
1	General Public Views and Tradeoff Priorities	The literature probes the question of what the general public values and prioritizes more, pro-health or pro-economy.	[13,37,38]
2	Public Health Costs of an Unchecked COVID-19	The literature addresses the consequences for public health if COVID-19 goes unchecked, ranging from morbidity to possible health system inundation.	[4,33,39,40,41]
3	Social Distancing as an Effective Strategy for Improving Public Health Metrics and Reducing Medical and Potential Economic Costs	Social distancing has been deemed an effective NPI strategy by experts for public health metrics, but some countries’ findings show otherwise. Models show future economic gains and saved medical expenses as a result of social distancing steps.	[34,40,41,42,43,44,45,46,47]
4	Is Social Distancing Justified given the Economic Toll?	It is accepted that the economic perils of a pandemic are not trivial; however, less agreement exists with regard to whether the consequences of various NPIs, such as social distancing, are appropriate. Some suggest that economic decline itself has an adverse effect on health, while others suggest that non-containment of the virus will naturally have a substantial economic toll. Several modeling studies have found that costs of social distancing far outweigh the benefits.	[33,34,44,45,48,49,50,51,52,53,54,55,56,57,58,59,60,61,62,63,64,65,66,67]
5	Gaging the Impact on the Global Economy	The literature extensively explores the impact of social distancing measures on the dislocation of the economy and its various sectors. Footprint of the pandemic has been and will be felt in both developing and developed countries, with those most vulnerable being most disproportionately impacted.	[32,33,42,51,52,58,64,65,66,68,69,70,71,72,73,74]
6	Cross-country Comparison	Various policies undertaken by countries are being considered and compared in terms of their impact on public health metrics and the economic toll. Countries’ policies range on a spectrum from limited measures to severe lockdowns. Findings suggest that the quick elimination of the virus and speedy reopening of the economy may be a way to balance between both health and economic objectives.	[59,66,75,76,77,78,79,80]
7	The Need for a More Nuanced Approach, the Role of Modeling for Optimization	A more nuanced approach to the HED is necessary in order to achieve optimal and effective policies which reconcile public health and economic interests. Several models present findings which aim to couple both domains, merging macroeconomic and epidemiological dynamics.	[33,43,59,73,80]
8	Proposed Viable Policy Intervention	Various proposed policy interventions have been recommended to contain COVID-19 and resolve the economic crisis. One such proposition includes achieving herd immunity among low-risk groups (such as among younger populations) to keep the economy running, while risk groups would be selectively quarantined. This approach is modeled and evaluated.	[36,46,47,50,51,53,56,59,60]

## Data Availability

Data sharing not applicable. No new data were created or analyzed in this study. Data sharing is not applicable to this article.

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
