# Peer review of "Deciphering the COVID-19 Health Economic Dilemma (HED): A Scoping Review"

_ijerph, 2021, doi:10.3390/ijerph18189555_

Round 1
Reviewer 1 Report
I have one observation for the Authors: why did you limit your analysis of the literature to 2020? Many other papers on the matter have been published in the following six months, written on the basis of additional expertise. Why not to include them too?
As far as the paper itself is concerned, I appreciate the objectives, the methodology, the results and in particular the conclusions, which express - from the amount of informations gathered, elaborated and compared - a list of reasonable suggestions for the planners both In the field of public health and in the field of economy. To try to eliminate the virus quickly while reopening the economic activities; not to focus only on solving the public health problems OR the economic problems, but to act simultaneously on both; not to privilege economy, because damages to public health would be terrible; to moderately restrict the mobility so that the flattening of the epidemiological curve and the reduction of the economic impact are obtained simultaneously. In conclusions, no draconian interventions but careful action on several fronts simultaneously
Author Response
We thank the reviewer for his comment. Please see below the delineated response to each of the comments:
I have one observation for the Authors: why did you limit your analysis of the literature to 2020? Many other papers on the matter have been published in the following six months, written on the basis of additional expertise. Why not to include them too?
Response: We completely agree with the reviewer that important contributions to the literature is a continuous process and must not be limited to the year 2020. Our analysis reflects the specific time-period of up to the end of 2020, when we conducted the analysis. Nonetheless, upon the time of completion of the review, we rapidly checked the literature once more and added some relevant articles that were published in the beginning of the year 2021 (see references 24, 59, 81, 84, 85).
As far as the paper itself is concerned, I appreciate the objectives, the methodology, the results and in particular the conclusions, which express - from the amount of informations gathered, elaborated and compared - a list of reasonable suggestions for the planners both In the field of public health and in the field of economy. To try to eliminate the virus quickly while reopening the economic activities; not to focus only on solving the public health problems OR the economic problems, but to act simultaneously on both; not to privilege economy, because damages to public health would be terrible; to moderately restrict the mobility so that the flattening of the epidemiological curve and the reduction of the economic impact are obtained simultaneously. In conclusions, no draconian interventions but careful action on several fronts simultaneously
Response: We thank the reviewer for this input and am glad that our findings strengthen the knowledge on this important topic
Reviewer 2 Report
The article is very clear and well structured. It gives an exhaustive framework of the literature reaching the goal of providing a valuable instrument for decision making facing the pandemic situation.
The only suggestion regards the methodology section in wich would be interesting to better underline the geographic provenance of the analyzed papers in order to correlate their focus with the situation in the different countries.
Such considerations could possibly be developed in the conclusions if there are any significant relationships between certain research attitudes and the Covid impact in different areas.
As reader, I would also be intrigued by the way time and events (and new findings) are going to impact the research path.
Since this literature review is obviously restricted in time, I doubt the authors can develop this issue in the framework of this article, but it could be a suggestion for future contributions.
Author Response
We thank the reviewer for his comment and have revised the manuscript accordingly. We feel that this version is indeed markedly improved. Please see below the delineated response to each of the comments:
The article is very clear and well structured. It gives an exhaustive framework of the literature reaching the goal of providing a valuable instrument for decision making facing the pandemic situation.
The only suggestion regards the methodology section in which would be interesting to better underline the geographic provenance of the analyzed papers in order to correlate their focus with the situation in the different countries. Such considerations could possibly be developed in the conclusions if there are any significant relationships between certain research attitudes and the Covid impact in different areas. As reader, I would also be intrigued by the way time and events (and new findings) are going to impact the research path. Since this literature review is obviously restricted in time, I doubt the authors can develop this issue in the framework of this article, but it could be a suggestion for future contributions.
Response:
We have added to the conclusions section a statement on potential future research inquiries which could examine geographic specific COVID-19 impact and respective attitudes regarding the Health Economic Dilemma.
Reviewer 3 Report
I find this work very interesting; one aspect to consider would be the increased expenditure due to the lack of attention to "non-COVID pathologies."
Worsening health due to long waiting lists is also a factor to be taken into account.
It seems that the measures recommended by the World Health Organization regarding distancing and hygiene measures are necessary to control the pandemic in the long term.
The economic cost should also be considered according to the lost lives of millions of people of productive age.
Author Response
We thank the reviewer for reviewing the manuscript and for his support.
As for the topic of "non-COVID pathologies", waiting times and lost lives of productive age - as we did not find these elements in the examined literature, we did not include this in the literature review. However, we have now included these elements on increased expenditure due to lack of attention to non-COVID pathologies and consideration of economic cost of productive age as additional elements that are recommended for further consideration. Please see the revised conclusions section.
Reviewer 4 Report
Dear colleagues, I've read with great interest your draft in which you have tried to shed light on the multifaceted issue of health economics regarding COVID-19 pandemic. This dramatic condition has forced many agents to take controversial decisions lacking of supporting background, and your paper is aimed to help to solve this later problem.
Your review summarizes the key messages from a selected search in the literature according to a well defined methodology, and provides a shallow overview of a number of (still) unanswered questions.
I missed a brief summary of the respective benefit of opposed opinions regarding antagonistic decisions, as such locking-down support or not, or prioritization for vaccination o clinical care in case of overwhelmed institutions, or many other in which care providers or regulators have to take very difficult decisions.
Otherwise, and in general terms, I feel that your paper needs no other major changes, and can be proposed to the editor for his/her consideration.
Typo: "laizzes" should be "laissez" in line 181.
Author Response
We thank the reviewer for his important comment. We have revised the manuscript accordingly. Please see specific response to each comment:
Concerning a brief summary of the respective benefit of opposed opinions regarding antagonistic decisions;
Response: such a summary has now been added to the manuscript. See section 3.9
Typo: "laizzes" should be "laissez" in line 181
Response: this was modified accordingly